# A prevalence study in Guadalajara, Mexico, comparing tuberculin skin test and QuantiFERON-TB Gold In-Tube

Arturo Plascencia Hernández[1,2], Rodrigo M. González Sánchez[1,2], Iván I. Hernández Cañaveral[1], Antonio Luévanos Velázquez[1,2], Pedro A. Martínez Arce[1,2], Alexander González Díaz[3], Manuel Sandoval Díaz[4], Yaxsier de Armas Rodríguez[3], Edilberto González Ochoa[3], Héctor Raúl Pérez Gómez[1] *

1 University Center for Health Sciences (Centro Universitario de Ciencias de la Salud), University of Guadalajara, Guadalajara, México, 2 "Fray Antonio Alcalde" Civil Hospital of Guadalajara, Guadalajara, México, 3 Pedro Kouri Institute of Tropical Medicine, Havana, Cuba, 4 Secretariat of Health, Guadalajara, Jalisco, Mexico

* hectorraul.perez@cucs.udg.mx

## Abstract

### Background

Tuberculosis (TB) is a prevalent disease throughout the world. The extent of TB illness in childhood is not clear; recent data shows that 10–20% of the cases are found in children under 15 years old. In 2017, 1 million children developed the disease, of which 9% were co-infected with HIV.

### Methods

A cross-sectional study that analyzed 48 children diagnosed with HIV-infection in Guadalajara, Mexico. The tuberculin skin test (TST) and QuantiFERON-TB Gold In-Tube test (QFT) were performed and compared to diagnose latent TB infection (LTBI).

### Results

The average age was 9 years old (± 4), with an age range of 1–16 years; the 6-12-year-old group predominated with 50% of cases. 27 patients (56%) were male; 83% had received the BCG vaccination and 23% had a history of being contacts of TB cases. In the study, 40 patients (83%) were without immunosuppression; seven (15%) with moderate immunosuppression, and only one patient had severe immunodeficiency. Overall, 3 of the 48 children (6.2%) had a positive TST, while 8 out of 48 (16.6%) had a positive QFT. The concordance between the two tests was 89.6% (43/48) with Kappa = 0.5 (95% CI, 0.14–0.85).

### Conclusions

The QFT test represents an opportunity in the diagnosis of LTBI, particularly in pediatric HIV- patients. This is the first study that compares the two tests (TST and QFT) in children with HIV-infection in Guadalajara, Mexico.

**Data Availability Statement:** All relevant data are within the manuscript and its Supporting Information files.

**Funding:** The funders had no role in study design, data collection and analysis, decision to publish, or preparation of the manuscript.

**Competing interests:** The authors have declared that no competing interests exist.

## Introduction

Tuberculosis (TB) is a prevalent disease throughout the world. The World Health Organization (WHO) estimates that a quarter of the world's population is infected with *Mycobacterium tuberculosis* (MTB). Although the worldwide extent of the TB illness in childhood is not clear, recent data shows that 10–20% of the cases are found in children under 15 years old [1]. Likewise, children have a higher risk of developing active TB: 43% in children under 1 year of age, 24% between 1–5 years of age and 15% between 5–15 years of age, and HIV infection in pediatric ages, could increase the incidence of TB by a factor of around 8, increasing with degree of immunosuppression [2, 3].

The impact of HIV on the burden of TB in children has been less defined than it has been for adults. An estimated 1 million children developed the disease and 205,000 die of TB-related causes each year, of which 9% are co-infected with HIV [1]. Eighty percent of these deaths occur in children $< 5$ years old, with the majority (96%) of deaths occurring among children who did not receive treatment [3, 4]. Within this population, the prevalence of co-infection falls within the range of $<5\%$ in industrialized countries, in contrast to $>50\%$ in some African countries with a high load. HIV-positive people (including children), have up to 20 times more risk of developing TB compared to the general population [1–3]. In addition, antiretroviral therapy in HIV-infected children, decrease the risk of TB by around 70% [3].

According to Secretariat of Health of Mexico, in 2019, 618 pulmonary and extrapulmonary TB cases were reported in childhood (less than 14 years old). According to estimates, the incidence is 3 (2.2–3.7/100, 000 inhabitants), and 5% of cases are associated with HIV-infection [1, 5].

TB diagnosis in pediatrics remains a challenge. The main impediments are the paucibacillary nature of the organism, the non-specific symptoms in the clinical picture (particularly in HIV infected) and difficulty in obtaining sputum samples; whereby diagnosis is only confirmed in 10–15% of positive sputum smears and close to 70% of negative cultures of probable cases, frequently with likely to rapidly progress to disseminated or extrapulmonary TB in the absence of appropriate treatment [2–4, 6]. Thereby, different kind of diagnostic test, has been developed in recent years, including molecular type (such as MTB-Xpert MTB-RIF assay) and serum based antigen, which has improved the sensitivity and specificity in the diagnosis of active TB in HIV and seronegative patients [7].

Until 2001, the tuberculin skin test (TST) was performed to diagnose latent TB infection (LTBI) [8]. In 2005, the Food and Drug Administration (FDA) authorized a new in vitro test, QuantiFERON-TB Gold, for the diagnosis of both LTBI and active TB. This test detects the amount of interferon gamma (IFN-γ) released by lymphocytes in the blood of sensitized people, using the Enzyme-Linked Immunosorbent Assay (ELISA). It contains mixtures of synthetic peptides similar to those presented by MTB and absent in the Bacillus Calmette Guerin (BCG) used for vaccination: white secretory early antigen (ESAT-6), culture-filtered protein (CFP-10) [9, 10] and in later years the TB 7.7, which reduces the number of indeterminate results and increases the diagnostic sensitivity and specificity [11–13]. Recently, to improve the sensitivity of the QFT-GIT in young children, or people with recent exposure, or with HIV infection, was developed the QFT-Plus that has one additional tube for the induction of cell-mediated immune responses from both CD4+ 31 and CD8+ T cells. Nevertheless, in a systematic review and meta-analysis of studies comparing the diagnostic performance of QFT-Plus to other tests for LTBI detection, no improvement in sensitivity and specificity was encountered with this assay [12]. The timely diagnosis of latent tuberculosis in populations at high risk of progression to active disease (such as in patients with HIV infection) is essential, because at least for now, international recommendations establish that in such a scenario the patient

should receive chemoprophylaxis to avoid it [6]. Also, in children it is mandatory to identify the prevalence of LTBI in HIV infected patients and other immunosuppressive states. This prevalence can be as low as 1% and up to 31.5% [8–10] and often depends on the burden of tuberculosis in a given region. Probably in the near future, some newly described tests will be able to differentiate between patients with latent infection and increased risk of progression on whom such chemoprophylaxis should be focused [14].

In international literature, only a few studies address the issue of LTBI in the pediatric population in Mexico, mainly in populations living in the border with United States [11, 15]. For this reason, the objective of the present study is to identify the prevalence and to evaluate the concordance of the TST and QFT diagnosis tests for LTBI in a group of children with HIV-infection treated in Guadalajara, Mexico, city located in the west-center of the country, where no previous studies has issued this question.

## Material and methods

48 children diagnosed with HIV-infection included in this cross-sectional study between May 2011 and April 2013 at the HIV/AIDS clinic of the "Fray Antonio Alcalde" Civil Hospital of Guadalajara (CHG-FAA) were analyzed.

The following patient data was collected: age; sex; history of contacts with TB cases; BCG vaccination, corroborated by the immunization record or scar on the arm; viral load, and CD4 + T lymphocyte levels. The original Classification System from the Centers for Disease Control and Prevention (CDC) was used to assess immune status, which classified children into: without immunosuppression, moderate immunosuppression and severe immunosuppression [14]. Taking into account the age groups used for this classification, the variable was stratified into similar groups: under 1 year of age, 1–5 years, 6–12 years, and $\geq$ 13 years.

A single 5 ml sample of blood was taken for the QFT test; aliquots of heparinized whole blood, incubated with MTB antigens, a negative control (nil) and a positive mitogenic control (phytohemagglutinin). After 24 to 36 hours of incubation at 37°C, the plasma concentration of IFN-γ was determined by ELISA. The test's veracity depends on the generation of an appropriate standard curve, which must cover the specific criteria established by the test. A positive result is considered when the amount of IFN- γ released in response to antigens ESAT-6, CFP 10 and TB 7.7 has a value greater than 0.35 IU/ml, according to the manufacturer's instructions [11–13].

The TST, consisting of a delayed hypersensitivity reaction that starts at 36 hours, after the intradermal application of Protein Purified Derivative in the forearm, reaching a maximum at 48–72 hours with induration at the site of inoculation, and positivity is measured in millimeters (mm). For HIV patients, a 5 mm induration is considered positive and indicates MTB infection [8].

For the quantitative variables, the mean and standard deviation and ranges were used, and for the qualitative ones, the frequency and percentages were determined. The Chi-square test (Fisher's Exact test, for samples less than five) and Mann Whitney U test were used to compare proportions and median, respectively. To calculate the concordance between the tests, Cohen's kappa coefficient (κ) was used.

### Exclusion criteria

Patients with active or past TB disease were excluded from the study.

### Ethical considerations

Because the investigation of LTBI (through TST and/or interferon gamma release assay, in HIV-patients, it's a quality standard of care procedure, and fundamental to decide the use of

chemoprophylaxis against active tuberculosis and the present study was based on the results of those tests, previously documented in the clinical records of the patients; it was sufficient to have the parents or guardian´s signature of the informed consent for general care of the minors, either as hospitalized or outpatient. Either way, this research was subjected to evaluation and authorized by the Committees of Research, Ethics and Biosafety both, of the Civil Hospital of Guadalajara and the Secretariat of Health of Jalisco Mexico.

## Results

Tables 1 and 2 shows the epidemiological profiles of the 48 children analyzed in the study, and the results of TST and QFT, according their main characteristics studied: age, gender, BCG vaccination, immune status, CD4 cell counts and viral load. The average age was 9 years (± 4 years), with a range of 1–16 years; the group of 6–12 years predominated with 50% followed by the ≥ 13 years' group with 31%. 27 patients (56%) were male; 83% had received the BCG vaccination, and 23% had a history of being contacts of TB cases.

In relation to the immunological status, 40 patients (83%) were without immunosuppression; seven (15%) with moderate immunosuppression, and only one patient had severe immunodeficiency. The mean CD4+ T lymphocyte levels were 1 041 cells/μL (± 526.33) with values between 12 and 2 752 cells/ μL, while the mean for the viral load was 23 587 copies/ μL (± 110 095), with values between 20 and 706 000 copies/ μL.

Overall, 3 of the 48 children (6.2%) had a positive TST, while 8 out of 48 (16.6%) had a positive QFT. The mean age of the three positive TST cases was 8.6 years (range 3–15), with one case in each age group. In the eight positive QFT cases, an average age of 9.2 years was determined (range 3–15). An increase in the diagnosis of LTBI was observed with the QFT, from

**Table 1. Results of TST and QFT test in the diagnosis of LTBI, according to age, gender, BCG vaccination and TB-contact.**

| VARIABLE | TOTAL | | Tuberculin Skin Test | | | | QuantiFERON-TB-Gold Test | | | | P value[*] |
|---|---|---|---|---|---|---|---|---|---|---|---|
| | | | Positive (n = 3) | | Negative (n = 45) | | Positive (n = 8) | | Negative (n = 40) | | 0.19 |
| | N | % | N | % | N | % | N | % | N | % | |
| AGE GROUP (YEARS) | | | | | | | | | | | |
| 1–5 | 9 | 18.8 | 1 | 11.1 | 8 | 88.9 | 2 | 22.2 | 7 | 77.8 | 1.0 |
| 6–12 | 24 | 50.0 | 1 | 4.2 | 23 | 95.8 | 3 | 12.5 | 21 | 87.5 | 0.6 |
| 13–16 | 15 | 31.3 | 1 | 6.7 | 14 | 93.3 | 3 | 20.0 | 12 | 80.0 | 0.59 |
| ALL | 48 | 100 | 3 | 6.25 | 45 | 93.7 | 8 | 16.6 | 40 | 83.3 | 0.76 |
| Age Mean (SD) Range | 9.0 (±4) 1–16 | | 8.66 (±6.02) 3–15 | | 9.4 (±3.9) 1–16 | | 9.2 (±4.6) 3–15 | | 9.4 (±3.97) 1–16 | | 0.86 |
| GENDER | | | | | | | | | | | |
| Female | 21 | 43.8 | 3 | 14.3 | 18 | 85.7 | 5 | 23.8 | 16 | 76.2 | 0.69 |
| Male | 27 | 56.3 | 0 | 0.0 | 27 | 100.0 | 3 | 11.1 | 24 | 88.9 | 0.06 |
| BCG VACCINATION | | | | | | | | | | | |
| Yes | 40 | 83.3 | 3 | 7.5 | 37 | 92.5 | 7 | 17.5 | 33 | 82.5 | 0.3 |
| No | 8 | 16.7 | 0 | 0.0 | 8 | 100.0 | 1 | 12.5 | 7 | 87.5 | 1 |
| TB CONTACT | | | | | | | | | | | |
| Yes | 11 | 22.9 | 1 | 9.1 | 10 | 90.9 | 2 | 18.2 | 9 | 81.8 | 1 |
| No | 37 | 77.1 | 2 | 5.4 | 35 | 94.6 | 6 | 16.2 | 31 | 83.8 | 0.26 |

[*]comparison test of measures between positive cases according to the test used for the total and sub-categories.

**Table 2. Results of TST and QFT test in the diagnosis of LTBI, according to immune status, CD4-cells counts and HIV-viral load.**

| VARIABLE | TOTAL | | Tuberculin Skin Test | | | | QuantiFERON-TB Gold test | | | | P Value* |
|---|---|---|---|---|---|---|---|---|---|---|---|
| | N | % | Positive (n = 3) | | Negative (n = 45) | | Positive (n = 8) | | Negative (n = 40) | | 0,19 |
| | | | N | % | N | % | N | % | N | % | |
| IMMUNE STATUS | | | | | | | | | | | |
| Without immunosuppression | 40 | 83.3 | 3 | 7.5 | 37 | 92.5 | 6 | 15.0 | 34 | 85.0 | 0.47 |
| Moderate immunosuppression | 7 | 14.6 | 0 | 0.0 | 7 | 100.0 | 1 | 14.3 | 6 | 85.7 | 1 |
| Severe immunosuppression | 1 | 2.1 | 0 | 0.0 | 1 | 100.0 | 1 | 100.0 | 0 | 0.0 | - |
| T CELLS CD4* LYMPHOCYTE LEVELS (CELLS/MM3) | | | | | | | | | | | |
| Average (SD) Range | 1 041 (± 526.33) 12–2752 cells/ mm³ | | 1060 (±179.9) 864–1217 | | 1040 (±542.6) 12–2752 | | 1 110.2 (±709.2) 12–2218 | | 1027.5 (±492.3) 340–2752 | | 0.78 |
| Median (IQR) | | | 1101 (864–1217) | | 1000 (659–1260) | | 982.5 (751–1592) | | 1006 (681–1234) | | |
| VIRAL LOAD* COPIES/ MM3 | | | | | | | | | | | |
| Average (SD) Range | 23 587 (± 110 095) 20–706000 | | 54 (± 24.2) 40–82 | | 25309 (± 113953) 20–706000 | | 27755.14 (± 73267.4) 40–193910 | | 22799 (± 116529.3) 20–706000 | | 0.82 |
| Median (IQR) | | | 40 (40–82) | | 40 (40–106) | | 67 (40–107) | | 40 (40–99) | | |

*comparison test of measures between positive cases according to the test used for the total and sub-categories.

11.1 to 22.2% in children of 1–5 years; from 4.2 to 12.5% in those aged 6–12 years, and from 6.7 to 20% in those over 13 years of age.

All positive TST cases were female, which represented 14.3% of the total gender group. With the QFT, 5 out of 21 (23.8%) of the females were positive and 3 of the 27 (11.1%) children male. Forty children in our casuistic (83.3%) were vaccinated with BCG, of which 3 (7.5%) and 7 (17.5%) were TST and QFT positive respectively. Out of the eight (16.7%) BCG-unvaccinated cases, none had a positive TST and only one had a positive QFT.

In relation to the history of contact with TB cases, the largest number of cases was found in the group that did not report any contact with TB cases (37 of 48 patients, 77.1%). In this subgroup, there were two and six positive cases for the TST and QFT, respectively. However, the positivity percentage was higher among contacts with values of 9.1% and 18.2% for the TST and QFT positivity, respectively.

Regarding the immune status, in the group without immunosuppression, three positive TST cases (7.5%) were found, and this group's positivity for the QFT doubled with 15% (6 out of 40). Additionally, two cases were detected with the QFT; one with moderate immunosuppression that represented 14.3% of this group, and the only case with severe immunosuppression.

The average CD4+ T lymphocytes in the TST positive cases was 1 060 (± 179.9) with values between 864 and 1 217 cells/ μL; while in the negative cases, the mean was 1 040 (± 542.6) with a range between 12 and 2 752 cells/ μL. In the QFT positive tests, the mean CD4+ T lymphocytes was 1 110 (± 709.2) with values between 12 and 2 218 cells/ μL; in the QFT negative cases the mean of CD4+ T lymphocytes was 1027.5 (±492.3) and rank of 340–2752.

Regarding viral load, the mean value was 54 copies/ μL (± 24.24), with a range between 40 and 82 copies/ μL for TST positive cases; while for negative cases, the values oscillated between 20 and 706 000, with an average of 25 309 (± 113 953). For the QFT positive cases, an average of 27 755 (± 73 267.4), with a range between 40 and 193 910 is reported; in the negative cases, the mean was 22 799 (± 116 529.3), with a range between 20 and 706 000 copies/μL.

In the present study, no significant differences were observed within the same groups, nor among the positive cases in both tests for any of the variables analyzed. In general, the concordance between the two tests was 89.6% (43/48), κ = 0.5 (95% CI,0.14–0.85).

**Table 3. Characteristics of patients with positive results to TST and/or QFT in the diagnosis of LTBI according to selected characteristics.**

| CASE | AGE IN YEARS | SEX | BCG | TB CONTACT | IMMUNOLOGICAL STATUS | CD4+ LEVELS | VIRAL LOAD | TEST RESULT | |
|------|--------------|-----|-----|------------|----------------------|-------------|------------|-------------|-----|
| | | | | | | | | TST | QFT |
| 1 | 3 | Fem | Yes | Yes | w/o immunosuppression | 1 217 | 82 | + | + |
| 2 | 4 | Male | Yes | Yes | Moderate immunosuppression | 863 | 40 | - | + |
| 3 | 6 | Fem | Yes | No | w/o immunosuppression | 2 218 | 67 | - | + |
| 4 | 8 | Fem | Yes | No | w/o immunosuppression | 1 101 | 40 | + | + |
| 5 | 11 | Fem | Yes | No | w/o immunosuppression | 1 967 | 107 | - | + |
| 6 | 13 | Male | Yes | No | w/o immunosuppression | 640 | ND* | - | + |
| 7 | 14 | Male | No | No | Severe immunosuppression | 12 | 193 910 | - | + |
| 8 | 15 | Fem | Yes | No | w/o immunosuppression | 864 | 40 | + | + |

* Not Detected.

Table 3 shows the concordance between both tests according to the characteristics analyzed in the study.

## Discussion

To our knowledge, this is the first study that compares the two tests (TST and QFT) in the diagnosis of LTBI in children with HIV-infection in Guadalajara, Mexico. The diagnosis of LTBI in the pediatric population with HIV-infection represents an opportunity to prevent imminent progression to active TB, due to the fact that children develop the disease more frequently than older patients. It is reported that the pediatric population with CD4+ T lymphocyte levels lower than <15% (severe immunodeficiency) have a higher risk of developing severe forms, relapses, infection from multiresistant strains, adverse reactions to antituberculous drugs and higher mortality [2, 16]. The probability of developing the disease varies from 2.5% to 15% per year, with a risk of 25 to 50 times higher than in people without HIV infection [2, 3, 6].

Recently, the relationship between the QFT and the diagnosis of LTBI and active TB in pediatric population has been demonstrated [6]. This is also the case for patients who present TST indurations greater than 15 mm, positive culture and those with lower risk of false positives due to BCG in children under 5 years [16–19].

Identifying the sensitivity and specificity of TST and IGRAs in the diagnosis of LTBI represents a great difficulty, because in that setting there is not a comparative "gold standard test"; on the other hand, it is easier to define them for active tuberculosis, by having more specific elements and benchmarking objectives such as MTB cultures or molecular tests such as the MTB-Xpert MTB-RIF assay, or the combination of a TB-compatible clinical picture with findings in imaging studies, positive Ziehl–Neelsen staining smears and/or histological studies. However, the sensitivity and specificity of both tests (TST and QFT) in the pediatric population has been compared in several investigations [6, 19–22]. The TST showed a lower sensitivity of 78% and a specificity of 49%, while the QFT test had a sensitivity of 89% and a specificity of 96% for LTBI. The latter has a lower percentage of false positives related to BCG vaccination and cross-reaction with other mycobacteria; better tolerability among children; greater speed and accuracy in the results [16–18, 20, 22]. In our study, we found a higher proportion of QFT positivity compared to TST, both in the general casuistry and in the different subgroups analyzed (age, history of BCG vaccination, contacts of active TB cases and immune status categories. However, it is possible that the sample size (48) did not allow us to identify statistical significance of these differences.

Another interesting aspect of this research is the way in which the immunological characteristics of patients with HIV were studied. It is reported that the QFT test in HIV immunocompromised patients shows a limited value due to a high number of indeterminate responses, sensitivity of 60% and specificity of 59%, which are lower compared to the results of these tests among the general population [16, 23]. TST response can also be reduced in this group of patients, which justifies our study. It was noted that a higher percentage of sensitivity and specificity in the QFT is achieved in patients with CD4+ T lymphocyte levels greater than 150–200 cells /μL [23].

However, it is important to note that in this study, 75% of the individuals analyzed did not possess immunosuppression; (3 out of 48) and (8 out of 48) were LTBI positive according to the TST and QFT, respectively. This result represents a possible epidemiological alert, since individuals without immunosuppression can become infected and become reservoirs for MTB. In addition, one patient with severe immunosuppression tested positive with the QFT; a result that is not significant, but nevertheless of interest, due to the usefulness of this test in this type of patients. We can confirm that false negatives for immunosuppression or concomitant diseases fall into an area in which more training and resources are needed.

Reports of comparative studies between TST and QFT tests in pediatric patients are generally discordant, as indicated by the kappa values (less than 0.6). These are higher in the population vaccinated with BCG, suggesting that at least some of this discordancy may be attributable to this factor [24]. In our study we did not identify any TST+/QFT- cases, even though 83.3% had a history of BCG vaccination; this can be explained by the fact that 81% of the children were over 6 years of age and all those vaccinated, received BCG in the first three months of life, which is consistent with the consideration that more than 5 years after vaccination a positive TST could be indicative of LTBI and less probably positive-false due to the vaccine. In fact, in a recent study carried out in India whose objective was to determine the concordance between QFT-GIT and TST in 33 children (without HIV) vaccinated with Bacillus Calmette-Guerin, a false positive rate of TST of 83.3% and 33.3% among children under and over 4 years of age was respectively estimated, which suggests that more years after vaccination, the lower the likelihood of false positive TST [25].

In Denmark, the concordance between both tests was good (87% and 89%), which was similar in the United Kingdom, reflecting a panorama of industrialized countries with a low disease burden [26, 27]. These studies reflect that the incidence of TB is perhaps an important concordance factor between TST and QFT. An interesting study that shows the results of 15 studies in these areas did not show statistically significant differences in the sensitivity between the of both TST (88.2%, 95% confidence interval [CI] 79.4–94.2%) and QFT (89.6%, 95% CI 79.7–95.7%) tests in immunocompetent children [28].

The immune status of the patient affects the relationship between both tests. Other studies involving HIV-negative individuals corroborate this hypothesis [24]. In addition, there have been documents that show a low concordance in immunosuppressed patients with autoimmune diseases [29]. In general, there is poor agreement between the TST and QFT tests in individuals with rheumatology diseases, considering the TB burden factor [30–32].

According to some studies, Mexico is considered a country with a high prevalence of LTBI [33] but that can vary in different areas within the country. Our study identified a prevalence of LTBI in HIV-infected children (under 16 years of age) of 16.6% based on QFT, in one of the largest teaching hospitals, located in the central-western region of the country and that was lower than that reported in a study in children with HIV infection living in the border region between Tijuana, Mexico and San Diego California, United States using TST and QTF-GIT, where a prevalence of LTBI of 20.3% was identified [11]. This could be a reflection of the

higher incidence of tuberculosis in the U.S.-Mexico border region, which accounts for 30% of the total TB cases recorded in both countries [34].

Comparative evaluation of IGRAs in the diagnosis of LTBI in children with HIV infection is certainly an area of interest. Some studies have shown high levels of agreement (93%, κ = 0.83) between QFT-IT and and an enzyme-linked immunospot assay (ELISPOT) in children at risk of LTBI or active tuberculosis [35]. A meta-analysis to assess different IGRAs (QFT-G, QFT-GIT and Enzyme Linked Immune absorbent spot [ELISPOT]) and TST found that the sensitivity of the three tests in children with active TB were similar. Pooled specificity was 100% for QFT-based tests and 90% for ELISPOT, but was much lower for TST (56% overall and 49% in children with BCG vaccination) [36].

Studies of this nature contribute to improving standards of care by adopting timely chemo-prophylaxis strategies (for example with isoniazid) and thus avoid progression to active tuberculosis.

The QFT test represents an opportunity in the diagnosis of LTBI, particularly in pediatric HIV-infected patients, in which the response mechanism to TST is diminished. This study highlights the high reliability of the QFT-Gold-IT test in the diagnosis of LTBI, despite low CD4+ T lymphocyte levels, in relation to symptomatic patients within close contact of tuberculous patients. This study argues that the periodic scrutiny of two tests in this population appears to contribute more to the identification of LTBI cases than using the TST alone.

Our study has some limitations. Although QFT was notoriously more sensitive than TST, the limited sample size (48 patients) probably influenced the finding of no statistical significance of the differences. Likewise, in a future study it would be interesting to compare different IGRAs such as QFT-GIT and Enzyme Linked Immune absorbent spot [ELISPOT] in the diagnosis of LTBI and active TB in children with HIV infection. It should be considered that our study was carried out in only one of the largest reference hospitals in Mexico and that could not fully reflect the situation in the country. A multicenter study in Mexico including several regions would be interesting. Finally, as with other studies, it is difficult to establish the sensitivity and specificity of these tests in the diagnosis of LTBI, since there is no comparative gold standard, as if there is one for active tuberculosis.

## Supporting information

**S1 Data.**
(XLS)

## Author Contributions

**Conceptualization:** Arturo Plascencia Hernández, Yaxsier de Armas Rodríguez, Edilberto González Ochoa, Héctor Raúl Pérez Gómez.

**Data curation:** Rodrigo M. González Sánchez, Iván I. Hernández Cañaveral, Antonio Luévanos Velázquez, Pedro A. Martínez Arce.

**Formal analysis:** Arturo Plascencia Hernández, Alexander González Díaz, Yaxsier de Armas Rodríguez.

**Investigation:** Rodrigo M. González Sánchez, Iván I. Hernández Cañaveral, Antonio Luévanos Velázquez, Pedro A. Martínez Arce, Manuel Sandoval Díaz, Edilberto González Ochoa.

**Supervision:** Arturo Plascencia Hernández, Antonio Luévanos Velázquez, Pedro A. Martínez Arce, Manuel Sandoval Díaz, Héctor Raúl Pérez Gómez.

**Validation:** Alexander González Díaz.

**Visualization:** Alexander González Díaz, Yaxsier de Armas Rodríguez, Edilberto González Ochoa.

**Writing – original draft:** Arturo Plascencia Hernández, Rodrigo M. González Sánchez, Yaxsier de Armas Rodríguez, Héctor Raúl Pérez Gómez.

**Writing – review & editing:** Arturo Plascencia Hernández, Yaxsier de Armas Rodríguez, Héctor Raúl Pérez Gómez.

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
