## [Decision Letter · Decision Letter 0]

25 Nov 2021

PONE-D-21-34415Latent tuberculosis infection in Mexican pediatric AIDS patients.

A prevalence study in Guadalajara, Mexico, comparing Tuberculin Skin Test and QuantiFERON-TB Gold In-Tube.PLOS ONE

Dear Dr. Perez-Gomez,

Thank you for submitting your manuscript to PLOS ONE. After careful consideration, we feel that it has merit but does not fully meet PLOS ONE’s publication criteria as it currently stands. Therefore, we invite you to submit a revised version of the manuscript that addresses the points raised during the review process.

We look forward to receiving your revised manuscript.

Kind regards,

Lei Gao

Academic Editor

PLOS ONE

Journal Requirements:

a) Did participants provide their written or verbal informed consent to participate in this study?

Additional Editor Comments

Please discuss the limitations of the study.

Please discuss why no TST+/QFT- was observed given most of the studied children were BCG vaccinated.

Reviewers' comments:

Reviewer's Responses to Questions

**Comments to the Author**

1. Is the manuscript technically sound, and do the data support the conclusions?

Reviewer #1: Yes

Reviewer #2: Yes

Reviewer #3: Yes

2. Has the statistical analysis been performed appropriately and rigorously? 

Reviewer #1: Yes

Reviewer #2: Yes

Reviewer #3: Yes

3. Have the authors made all data underlying the findings in their manuscript fully available?

Reviewer #1: Yes

Reviewer #2: Yes

Reviewer #3: Yes

4. Is the manuscript presented in an intelligible fashion and written in standard English?

Reviewer #1: Yes

Reviewer #2: Yes

Reviewer #3: Yes

5. Review Comments to the Author

Reviewer #1: This study showed a scientific flow in its context, using suitable epidemiological model and correct statistical testing. It showed that IGRA test can label positive cases more than TST but a constrain is that cannot know which is right as there is no gold standard test is available yet.

Reviewer #2: The study is relevant and well conducted. Analysis and presentation are simple, straight, and crisp. Two suggestions are noted for the consideration of authors.

The inclusion and exclusion criteria adopted by the authors may be mentioned.

AIDS patient, HIV patient are some of the nomenclatures used interchangeably by the authors. It seems the 40 (80%) of the study subjects without immunosuppression are also classified as AIDS. The authors may consider refining the text with the standard nomenclature CLHIV/PLHIV or HIV positive for better inclusion and clarity.

Reviewer #3: It's an interesting work. It coyuld be more interesting if the analysis used not only QuantiFeron test but also the ELISPOT test and comparing both results; data comparing results with both technics are very important as they show that ELUSPOT is better the QUANTIFeronfor the diagnosis of LTBI. As data are of a Guadalajara in Mexico, do you know a similar data in another location of Mexico?. May be interesting if it could compare similar data in the country.

6. PLOS authors have the option to publish the peer review history of their article (what does this mean?). If published, this will include your full peer review and any attached files.

Reviewer #1: **Yes: **Layth Al-Salihi

Reviewer #2: **Yes: **Shibu Balakrishnan

Reviewer #3: No

---

## [Author Response · Author response to Decision Letter 0]

17 Jan 2022

Reviewer #1, Professor Layth Al-Salihi:

 This study showed a scientific flow in its context, using suitable epidemiological model and correct statistical testing. It showed that IGRA test can label positive cases more than TST but a constrain is that cannot know which is right as there is no gold standard test is available yet.

 ANSWER: Fully agree with your observation; in the new revised manuscript we have decided to add the following paragraph: “Identifying the sensitivity and specificity of TST and IGRAs in the diagnosis of LTBI represents a great difficulty, because in that setting there is not a comparative "gold standard test"; on the other hand, it is easier to define them for active tuberculosis, by having more specific elements and benchmarking objectives such as MTB cultures or molecular tests such as the MTB-Xpert MTB-RIF assay, or the combination of a TB-compatible clinical picture with findings in imaging studies, positive Ziehl–Neelsen staining smears and/or histological studies.”

Reviewer #2 Professor Shibu Balakrishnan: 

The study is relevant and well conducted...…. Two suggestions are noted for the consideration of authors.

The inclusion and exclusion criteria adopted by the authors may be mentioned.

ANSWER: Fully agree with your observation; in the new revised manuscript we have decided to add the following paragraph: “Exclusion criteria. Patients with active or past TB disease were excluded from the study”.

AIDS patient, HIV patient are some of the nomenclatures used interchangeably by the authors…...

ANSWER: Totally according to your observation, in the new revised manuscript we have decided to use the terms "HIV-infected", "HIV-infection", "HIV-patients" instead of AIDS or AIDS patients.

Reviewer #3: 

It's an interesting work. It could be more interesting if the analysis used not only QuantiFeron test but also the ELISPOT test and comparing both results; data comparing results with both technics are very important as they show that ELUSPOT is better the QUANTIFeron for the diagnosis of LTBI. 

ANSWER: According to your valuable observation, in the new revised manuscript we have added two paragraphs: “Comparative evaluation of IGRAs in the diagnosis of LTBI in children with HIV infection is certainly an area of interest. Some studies have shown high levels of agreement (93%, κ = 0.83) between QFT-IT and and an enzyme-linked immunospot assay (ELISPOT) in children at risk of LTBI or active tuberculosis [35]. A meta-analysis to assess different IGRAs (QFT-G, QFT-GIT and Enzyme Linked Immune absorbent spot [ELISPOT]) and TST found that the sensitivity of the three tests in children with active TB were similar. Pooled specificity was 100% for QFT-based tests and 90% for ELISPOT, but was much lower for TST (56% overall and 49% in children with BCG vaccination). [36]” and “Likewise, in a future study it would be interesting to compare different IGRAs such as QFT-GIT and Enzyme Linked Immune absorbent spot [ELISPOT] in the diagnosis of LTBI and active TB in children with HIV infection.”

As data are of a Guadalajara in Mexico, do you know a similar data in another location of Mexico? May be interesting if it could compare similar data in the country.

ANSWER: According to your observation, in the new revised manuscript we have added this paragraph: “Our study identified a prevalence of LTBI in HIV-infected children (under 16 years of age) of 16.6% based on QFT, in one of the largest teaching hospitals, located in the central-western region of the country and that was lower than that reported in a study in children with HIV infection living in the border region between Tijuana, Mexico and San Diego California, United States using TST and QTF-GIT, where a prevalence of LTBI of 20.3% was identified [11]. This could be a reflection of the higher incidence of tuberculosis in the U.S.-Mexico border region, which accounts for 30% of the total TB cases recorded in both countries [33].”

Respectfully.

Héctor Raúl Pérez Gómez PhD. Corresponding autor.

---

## [Decision Letter · Decision Letter 1]

22 Feb 2022

Latent tuberculosis infection in Mexican pediatric HIV- patients.

A prevalence study in Guadalajara, Mexico, comparing Tuberculin Skin Test and QuantiFERON-TB Gold In-Tube.

PONE-D-21-34415R1

Dear Dr. Perez-Gomez,

We’re pleased to inform you that your manuscript has been judged scientifically suitable for publication and will be formally accepted for publication once it meets all outstanding technical requirements.

Kind regards,

Lei Gao

Academic Editor

PLOS ONE

Additional Editor Comments (optional):

Reviewers' comments:

Reviewer's Responses to Questions

**Comments to the Author**

1. If the authors have adequately addressed your comments raised in a previous round of review and you feel that this manuscript is now acceptable for publication, you may indicate that here to bypass the “Comments to the Author” section, enter your conflict of interest statement in the “Confidential to Editor” section, and submit your "Accept" recommendation.

Reviewer #1: All comments have been addressed

Reviewer #2: All comments have been addressed

Reviewer #3: All comments have been addressed

2. Is the manuscript technically sound, and do the data support the conclusions?

Reviewer #1: Yes

Reviewer #2: Yes

Reviewer #3: Yes

3. Has the statistical analysis been performed appropriately and rigorously? 

Reviewer #1: Yes

Reviewer #2: Yes

Reviewer #3: Yes

4. Have the authors made all data underlying the findings in their manuscript fully available?

Reviewer #1: Yes

Reviewer #2: Yes

Reviewer #3: Yes

5. Is the manuscript presented in an intelligible fashion and written in standard English?

Reviewer #1: Yes

Reviewer #2: Yes

Reviewer #3: Yes

6. Review Comments to the Author

Reviewer #1: The research team used a scientific flow of research work presentation. The methods used complies with run standards and norms. The article is amended as required.

Reviewer #2: The concerns I have raised during the review have been addressed. However, I would continue to advocate to consider careful use of the word 'patient' in an HIV context and use the standard global nomenclature 'People Living with HIV' and 'Children Living with HIV'. This may help in minimising the potential stigma and discrimination associated with HIV.

Reviewer #3: The manuscript is good, well related and analyzed. But.... se think is good for to be publised in your country, but not for international publications, because the number of subjects is low and there are a lot of studies on this relation of HIB and TB infection and the value of the IFN-gamma techniques . The study is good for to be published in your country.

7. PLOS authors have the option to publish the peer review history of their article (what does this mean?). If published, this will include your full peer review and any attached files.

Reviewer #1: **Yes: **Layth Al-Salihi

Reviewer #2: **Yes: **Shibu Balakrishnan

Reviewer #3: No

---

## [Editor Report · Acceptance letter]

2 Mar 2022

PONE-D-21-34415R1 

A prevalence study in Guadalajara, Mexico, comparing Tuberculin Skin Test and QuantiFERON-TB Gold In-Tube. 

Dear Dr. Pérez Gómez:

I'm pleased to inform you that your manuscript has been deemed suitable for publication in PLOS ONE. Congratulations! Your manuscript is now with our production department. 

Kind regards, 

on behalf of

Dr. Lei Gao 

Academic Editor

PLOS ONE